# A Robotic Recording and Playback Platform for Training Surgeons and Learning Autonomous Behaviors Using the da Vinci Surgical System

**Abhilash Pandya \*, Shahab Eslamian, Hao Ying, Matthew Nokleby and Luke A. Reisner**

Department of Electrical and Computer Engineering, Wayne State University, Detroit, MI 48202, USA; shahab.eslamian@wayne.edu (S.E.); hao.ying@wayne.edu (H.Y.); matthew.nokleby@gmail.com (M.N.); lreisner@wayne.edu (L.A.R.)

**\*** Correspondence: apandya@ece.eng.wayne.edu; Tel.: +1-313-577-9921

**Abstract:** This paper describes a recording and playback system developed using a da Vinci Standard Surgical System and research kit. The system records stereo laparoscopic videos, robot arm joint angles, and surgeon–console interactions in a synchronized manner. A user can then, on-demand and at adjustable speeds, watch stereo videos and feel recorded movements on the hand controllers of entire procedures or sub procedures. Currently, there is no reported comprehensive ability to capture expert surgeon movements and insights and reproduce them on hardware directly. This system has important applications in several areas: (1) training of surgeons, (2) collection of learning data for the development of advanced control algorithms and intelligent autonomous behaviors, and (3) use as a "black box" for retrospective error analysis. We show a prototype of such an immersive system on a clinically-relevant platform along with its recording and playback fidelity. Lastly, we convey possible research avenues to create better systems for training and assisting robotic surgeons.

**Keywords:** robotic surgery; surgical training; machine learning; surgical automation

---

## 1. Introduction

Surgery is a complex process and requires years of training. Teaching this complexity to people (or machines) is difficult and requires innovative methods. Conventional methods of laparoscopic and robotic surgical training often involve watching videos of surgeries, practicing on models/simulators, utilizing operating room time to practice on animals, and finally, clinical application under expert supervision on humans [1]. Currently, there are no comprehensive ways to capture and replay expert surgeon movements and insights on difficult procedures directly. Intuitive Surgical Inc. (the maker of the da Vinci system) has a system in place for recording data [2], but it doesn't currently support playback. Using that system, there has been work on capturing data for the objective assessment of robotic surgery training [3]. However, no system for direct training applications yet exists. We have designed a preliminary system for recording tasks or complete surgical procedures and replaying them on the hardware. Such a system could potentially assist in analysis/review, planning, automation, and training of surgery.

In this paper, we demonstrate a system capable of recording all hand-controller movements, robot arm motions, video, and other user inputs, which can be played back in a synchronized manner. Our main contribution is that we provide a detailed investigation of the fidelity of the reproduced movements and show how such a system can be tuned to produce more accurate results.

The system described herein allows major/complex operations to be recorded and to be kinesthetically felt and watched multiple times. With future work, the system has the potential to

provide an immersive haptic interface, augmented reality annotations on the video streams, and audio explanations. This could provide a rich set of knowledge-based surgery documentation and engage a trainee to better acquire the best practices of a surgical specialty from real surgeries. The trainee would be immersed in the surgery recording with sight, sound, and feeling, along with annotated knowledge.

Figure 1 shows the envisioned use of the playback portion of our system. In this paper, we focus on playing back the pre-recorded movement data and simultaneously displaying the pre-recorded video data. Other features like audio, graphical annotation, and clutch feedback are planned to be implemented in the future.

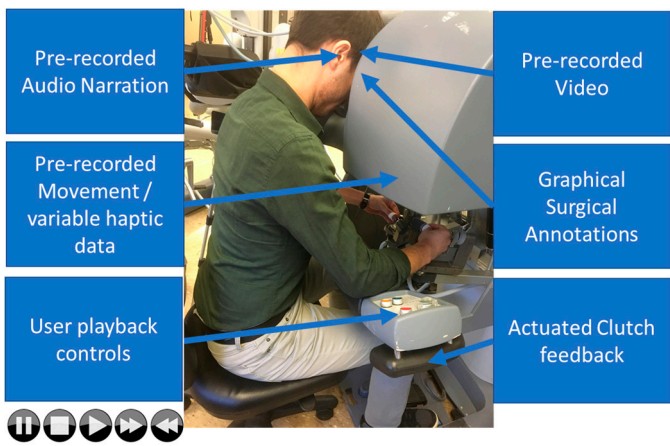

**Figure 1.** An envisioned immersive teaching environment using real surgical data. The trainee would be immersed in video, audio, and haptics to allow him to synchronously see, hear, and feel the pre-recorded surgery. It would also enable searching the recordings and adjusting the rate of playback.

Figure 5 shows the details of our recording portion. The same recording system could be of immense value in surgical automation. By allowing the creation of databases of recorded surgical procedures, the system would provide training inputs for machine learning algorithms that attempt to automate procedures (see Figure 2).

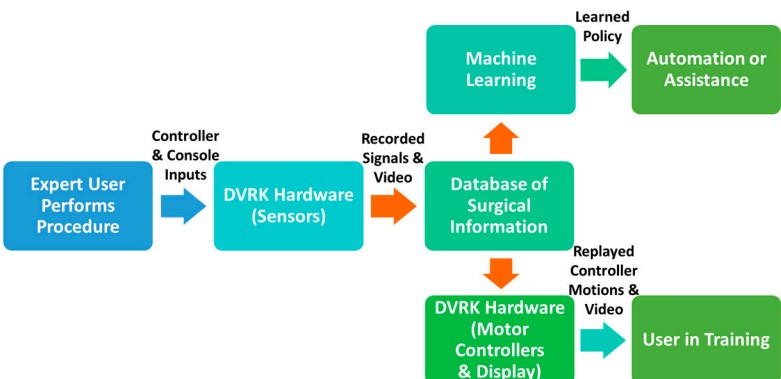

**Figure 2.** An overview of the recording and playback system enabled by the da Vinci Research Kit (DVRK), which allows data capture and playback for the envisioned machine learning and training applications.

## 2. Literature Survey

This system has many potential important applications, including (1) training of surgeons and (2) generation of training data for machine learning algorithms to automate certain tasks (see Figure 3). Data from an expert surgeon interacting with the console and hand controllers can be recorded in

a time-synchronized database of information. This data could be used to either learn policies from human demonstration or be replayed for use in training.

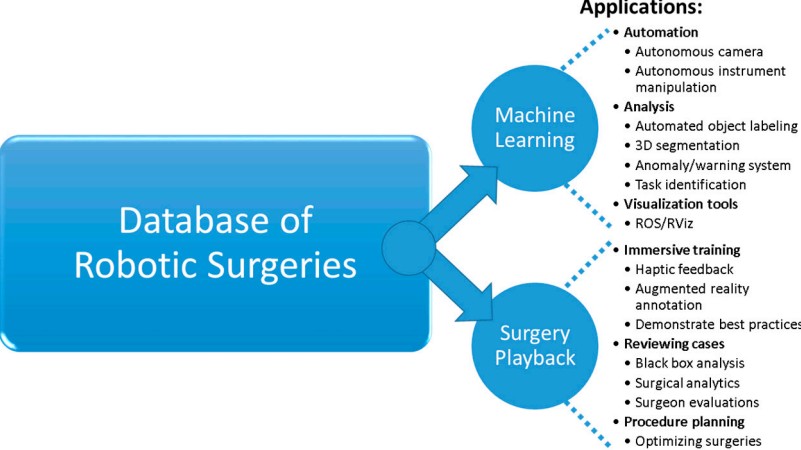

**Figure 3.** Applications that are enabled by a recording and playback system are shown on the right. They include automation of tasks, immersive training, and procedure planning.

## 2.1. Current Methods of Surgical Training and Evaluation

The field of surgical robotics training is of utmost importance [4]. Surgical robots are a relatively novel surgical platform, and new procedures that need meticulous dissemination continue to be developed for these systems. According to several recent robotic surgery training articles, the current state-of-the-art includes primarily three components: (1) didactics materials, (2) simulated robotics training environments, and (3) training with an expert. Currently, training is performed by watching videos of surgeries [5], practicing on models/simulators, utilizing operating room time to practice on animals, and finally clinical application under expert supervision on humans. The delicate movements needed for surgery are learned over time and many practice sessions [6–8].

This method has been followed for most training protocols. It is reliant on expert mentorship for surgical procedures. Unfortunately, if these experts retire and their methods are not fully recorded, the experience can be lost. It is also challenging to capture and learn all the nuanced movements and intricate knowledge required for surgery. Unlike any previous surgical platforms, robotic surgery offers the inherent ability for detailed recording of all aspects (hand controllers, robot arms, video, forces, velocities etc.) of a procedure that can be potentially used for training and other purposes.

Haptic interfaces for training have been used with virtual fixtures for laparoscopic training [9,10] and even in teaching complex calligraphy [11]. However, there is very little literature on the role of haptics/immersion in the area of robot-assisted endoscopic surgical training. Currently there is no comprehensive ability to capture expert surgeon movements and insights on difficult procedures directly. Advanced simulators are not totally realistic and lack expert feedback information from real recordings of procedures by skilled surgeons. Even the Fundamentals of Laparoscopic Surgery (FLS) program, which is commonly used to train laparoscopic surgeons, only provides a rudimentary evaluation framework based primarily on speed, accuracy, and subjective evaluations. To the best of our knowledge, there is no formal training method to help a surgeon improve their skills by simultaneously seeing and feeling the movements necessary for a procedure.

Our group previously demonstrated a basic recording and playback capability for the ZEUS Surgical System (a precursor to the da Vinci, also made by Intuitive Surgical Inc.) [12]. A simple system based on an Arduino microcontroller and a non-surgical robotic arm (built using six servomotors) also showed the concept of recording and playback [13]. Our system extends this work to a clinically relevant platform with multiple input/output devices, utilizes advanced 3D simulation software,

is more generalizable to other robots/systems, and evaluates both the recording and playback of the system.

### 2.2. Data for Automation of Surgical Procedures Using Machine Learning

To propel the field of surgical robotics to the next level, these systems must move from the domain of master-slave systems to the realm of intelligent assistants. They must be able to react to changing scenes, be in-tune with the procedure steps, and become better partners with the surgeon.

Recently, there has been an explosion in the performance and widespread adoption of machine learning tools in other fields. Much of this explosion has been fueled by deep learning which has advanced the state of the art on challenging tasks such as image recognition, automatic translation, and reinforcement learning. Recent works in *imitation learning* and *learning from demonstration* have used deep learning to enable machines to learn new, complex procedures from a limited number of human demonstrations [14–16]. Such methods can and should be leveraged for surgical robotics.

However, deep learning typically depends on large datasets for learning, and an extensive dataset of surgical robotics operations does not exist. Movement data for a limited set of bench-top tests are available [17] for research use, but the data is limited to a few examples and is not sufficiently large for deep learning. Any serious attempt at deep machine learning for complex surgical tasks requires a large amount of registered, time-synchronized, and high quality surgical data, as pointed out by Satava and others [18–21].

We contend that the development and verification of safe, clinically-useful intelligent systems will require such a dataset. In addition, any approach to formulate useful and safe autonomous systems will need an immense set of ground-truth data taken from actual surgeries to verify their performance.

If an immense set of surgical data could be collected and disseminated to the research community, it would open research avenues not only in automation and training, but also in other areas, such as visualization, procedure planning, and task identification among others (see Figure 3). The system described is a necessary step towards this goal.

## 3. Materials and Methods

### 3.1. da Vinci Surgical System and da Vinci Research Kit (DVRK)

Our research laboratory has a da Vinci Standard Surgical System with a da Vinci Research Kit (DVRK) interface [22], as shown in Figure 4. Also shown is a software simulation of our da Vinci test platform that is used for algorithm prototyping and the playback/visualization of the recorded data [23]. The DVRK is a hardware/software platform that helps researchers implement their ideas using a da Vinci Standard Surgical System. It is not for clinical use. Using the DVRK, we have full access to read and control the robotic arms of our da Vinci system. We can track and record the pose and movement data of the robotic arms (instruments and camera) using kinematic feedback of the DVRK.

The DVRK uses hardware control boxes (containing FPGA boards and amplifiers) and open software to enable computerized control of the robotic arms. This software extensively utilizes the popular open source Robot Operating System (ROS) framework [24], which helps facilitate the dissemination of developed software across the surgical robotics community. We use a subsystem, ROS bags, that allows time-synchronized recording of published data streams.

### 3.2. Robot Operating System Recording Software

The recorded data includes synchronized stereo video, kinematic data from both surgeon controllers, kinematic data from all arms, clutch activations, and system interface interactions. Everything is synchronized with timestamps. Figure 5 shows an example from our recording and playback system. This equipment, combined with the Robot Operating System (ROS) software framework [24], is used to record high resolution data at a high rate (100 Hz for encoders and

approximately 30 Hz for video) while participants are performing the selected tasks. In addition, there are very specific registration and calibration requirements of the system to accurately record and move the robot, which we describe in [25]. Briefly, using optimization-based techniques and paired-point matching between the camera arm and the tool arms, an accurate representation of the base transforms can be derived.

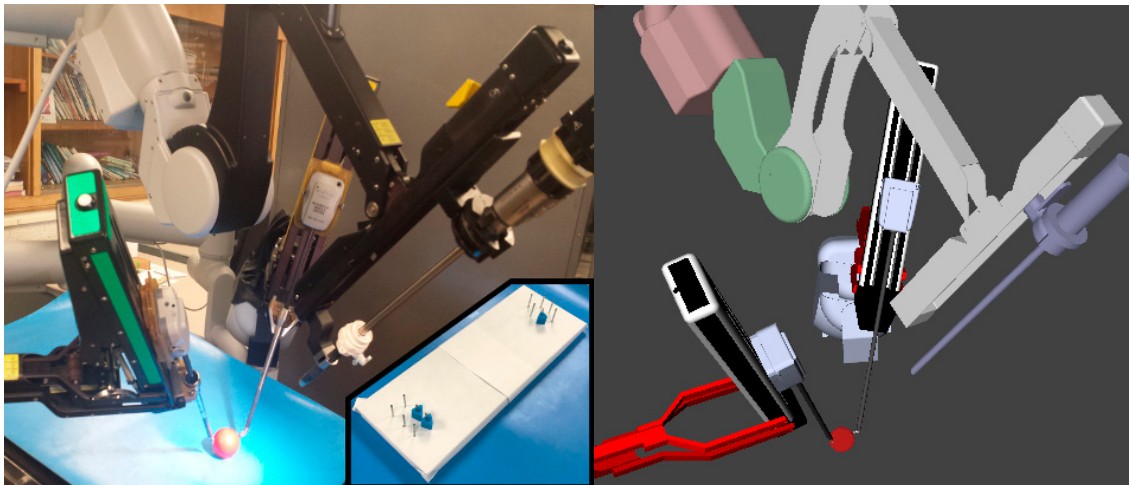

**Figure 4.** (**Left**) Our da Vinci Surgical System, which is used as a test platform for algorithm implementation and subject testing. (**Left**, **inset**) The modified peg transfer task. (**Right**) Our software simulation of the da Vinci test platform, which is used for algorithm prototyping and data playback/visualization. The simulated robot closely matches the real one, allowing rapid development and testing to be done first in simulation.

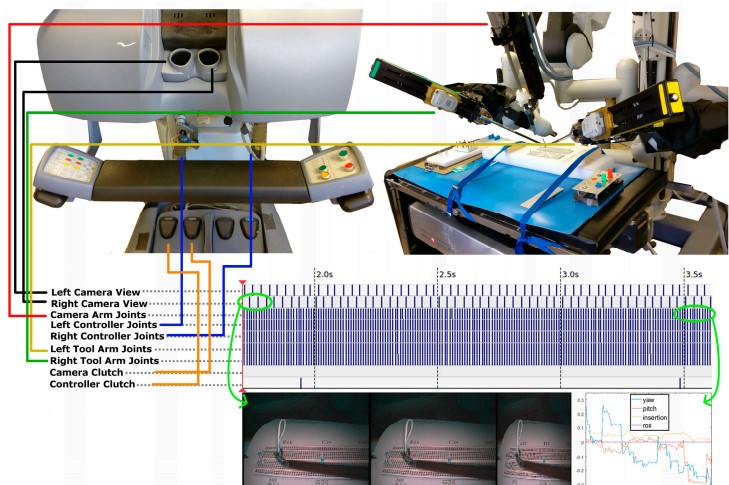

**Figure 5.** The recorded data will include stereo videos, kinematic data from both surgeon controllers, and kinematic data from all instrument arms, amongst other data. Everything will be synchronized with timestamps. This is an example from the recording and playback on a da Vinci Standard Surgical System.

### 3.3. Optimization of Playback Accuracy

During the recording phase, the data was simply collected from the encoders of the hand controllers. However, during playback the data is first sent to a PID (proportional–integral–derivative) control system containing 14 PID controllers. The PID control system monitors the error between the desired and commanded joint angles and tries to eliminate the error, causing the robot arm to move. If the PID gain values are not tuned properly, it results in poor transient performance and steady

state error upon replay of the data. Here we show how proper tuning of the PID gains leads to better replay accuracy.

### 3.3.1. System of Coupled Joints

There were seven joints for the right and left robot arms, each of which was controlled by a PID controller. The joints are considered mechanically symmetrical with respect to the middle line. The joints included the outer yaw, shoulder pitch, elbow pitch, wrist pitch, wrist yaw, wrist roll, and wrist platform (see Figure 6). Every joint has an encoder to measure each joint's movement, providing a real-time feedback signal for the PID controller. Because movements of the joints are mechanically linked, each arm is considered as a seven-input seven-output system with coupling from the standpoint of control theory. Ideally, one multi-input multi-output (MIMO) controller should be used to control such a MIMO system to achieve the best control performance. This is because in principle a MIMO controller can be designed to effectively handle the coupling, which is the key to good control performance. However, a MIMO mathematical model is necessary for such a controller design.

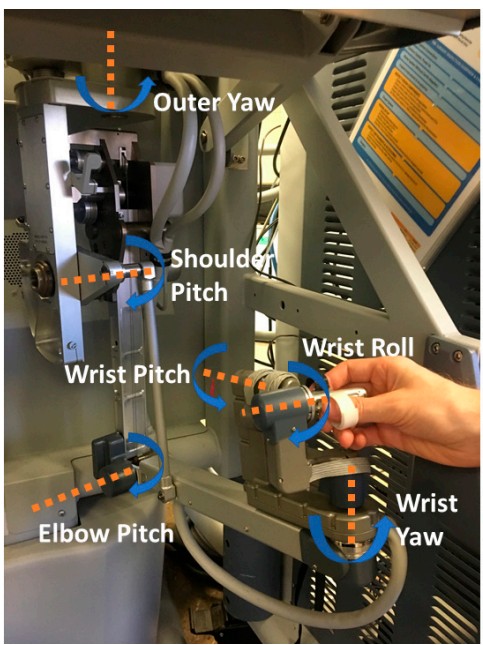

**Figure 6.** The relevant joints of the right hand controller. The left hand controller is symmetrical to this.

In our case, an accurate MIMO mathematical model for the arms is very difficult to obtain. Alternatively, the system is treated simplistically as being not coupled or only mildly coupled and thus this complexity can be ignored. This is a common practice in control theory. As a result, each joint is controlled independently by a nonlinear PID controller:

$$U(t) = \begin{cases} \left( K_\text{p}e + K_\text{i} \int e\, dt + K_\text{d} \frac{de}{dt} \right) \frac{|e|}{K_\text{n}} & \text{if } 0 < K_\text{n} < |e| \\ K_\text{p}e + K_\text{i} \int e\, dt + K_\text{d} \frac{de}{dt} & \text{otherwise} \end{cases} \tag{1}$$

where $K_\text{p}$, $K_\text{i}$, and $K_\text{d}$ are the proportional gain, integral gain, and derivative gain, respectively. $K_\text{n}$ is used to create a dead band when the joint is close to the goal. This improves the stability of the joints and is used because the encoder resolution on some of the joints are low, which makes it hard to dampen the controller using a higher $K_\text{d}$. This control algorithm came with the DVRK's software, and we did not modify it. We only tuned the three gains to improve its performance. The biggest advantage of a PID controller over other controllers in the literature is that its control gains have clear

meanings and can be manually and intuitively tuned to gain good control performance without a system's mathematical model.

The initial gains of the controllers (see Table 1) were taken from the DVRK software repository [26] (share/sawControllersPID-MTML.xml from commit e529c81d3179047c1b1f4b779727a7947017db18), which were good starting values. Note that the gains for the same joint of the left and right joints are identical, as expected. Also note that the integral gain for the wrist pitch, wrist yaw, and wrist roll is 0, meaning there is no integral control action. Lack of the integral gain can cause steady-state error of system response, degrading tracking performance.

**Table 1.** The gains of the seven PID controllers before our tuning was carried out [26]. The gains are identical for the right and left hand controllers.

| Joint | Proportional Gain $K_p$ | Integral Gain $K_i$ | Derivative Gain $K_d$ | Nonlinear Coeff. $K_n$ |
|---|---|---|---|---|
| Outer yaw | 30 | 1 | 1.5 | 0 |
| Shoulder pitch | 30 | 1 | 1.5 | 0 |
| Elbow pitch | 30 | 1 | 1.5 | 0 |
| Wrist pitch | 20 | 0 | 0.4 | 0 |
| Wrist yaw | 10 | 0 | 0.3 | 0.35 |
| Wrist roll | 1.2 | 0 | 0.04 | 0.35 |
| Wrist platform | 2 | 0.5 | 0.15 | 1 |

### 3.3.2. Tuning of the PID Control System

To ensure the tracking accuracies of the joints, we checked and manually fine-tuned the three control gains of each of the seven PID controllers (the initial gains are identical for the right and left controllers) to obtain the best control performance. We did so without relying on the mathematical models of the individual joints. Obtaining accurate models is very challenging and time consuming. Using the system identification technique to come up with an accurate MIMO mathematical model for the entire arm is almost impossible.

A step function was applied, and the observed rise time and overshoot of a joint's response were used to guide the gain-tuning process. The goal was to have small amounts of rise time, overshoot, and settling time as well as zero steady-state error in the tracking response for each joint. Initially, we tuned the gains of each controller individually to attain the best tracking response for that particular joint. When this was completed, the finger of the arm exhibited small vibration during the tracking test of the entire arm, indicating instability stemmed from one or more of the PID controllers of the arm. The root cause of this instability was the system's mechanical coupling.

We then changed our tuning strategy by tuning the gains of a joint controller first and then immediately testing the tracking performance of the whole arm to ensure the optimal response of each individual joint and the entire arm. The final gain-tuning results are presented in Table 2.

**Table 2.** The gains of the seven PID controllers for both the hand controllers after our tuning was performed. If a gain for a joint of the right arm differs from that for the same joint of the left arm, the right arm value is shown in parentheses.

| Joint | Proportional Gain $K_p$ | Integral Gain $K_i$ | Derivative Gain $K_d$ | Nonlinear Coeff. $K_n$ |
|---|---|---|---|---|
| Outer yaw | 39 | 1 | 5 | 0 |
| Shoulder pitch | 1 | 6 | 5.8 | 0 |
| Elbow pitch | 5 (3) | 4.6 (3.6) | 4 | 0 |
| Wrist pitch | 10 | 0.06 | 0.7 | 0 |
| Wrist yaw | 10 | 0 | 0.3 | 0.35 |
| Wrist roll | 1.2 | 0.016 | 0.04 | 0.35 |
| Wrist platform | 2 | 0.5 | 0.15 | 1 |

According to Table 1, there was no integral control for three of the seven joints, namely wrist pitch, wrist yaw, and wrist roll in the original PID controllers. Table 2 indicates that we were able to

add it to two of them for both arms (wrist pitch, and wrist roll). The effort of adding it to the wrist yaw joints led to a too small of stability margin, making us abandon it.

According to Table 2, the elbow pitch controllers for the two arms are not identical because their proportional-gains and integral-gains are different. These gain values produced the best responses. When the gains were the same for the two arms, the observed responses were quite different. The likely cause was small asymmetry between the two arms in our particular system.

### 3.4. Evaluation of System Accuracy

To evaluate the system, it is necessary to determine the fidelity of the recorded data and the accuracy of the system's playback. To accomplish this goal, we used the system to record a human operator moving a hand controller (with both slow and fast movements), and we analyzed the system's ability to play back the recorded hand controller movements under a variety of conditions (with a user's hands in the controllers, with two different sets of PID parameters, etc.). We also captured the movements with an external (optical) tracking system to assist in the evaluation. We will share the test datasets herein and a few task-oriented datasets on request to interested researchers.

In the following subsections, we first describe how we collected test data for our system. Then we explain how the test data was processed and analyzed. The results of the analyses are found in the Results section.

#### 3.4.1. Collection of Test Data

A total of 10 recordings (2 groups of 5 recordings) were collected for the evaluation of the system's accuracy. Each recording consisted of about 30 seconds of tracking data that captured the position of the left hand controller (also known as a master tool manipulator, or MTM). For the first group of recordings, the hand controller was moved at a slow speed (average 177 mm/s) intended to correlate to fine motions. For the second group, the hand controller was moved at a faster speed (average 372 mm/s) intended to correlate to gross translational motions. For both groups, the hand controller was moved through its entire range of motion in the *x*-axis, *y*-axis, and *z*-axis. The movements included brief pauses and changes in direction. Figure 7 shows a diagram of the recordings that were made.

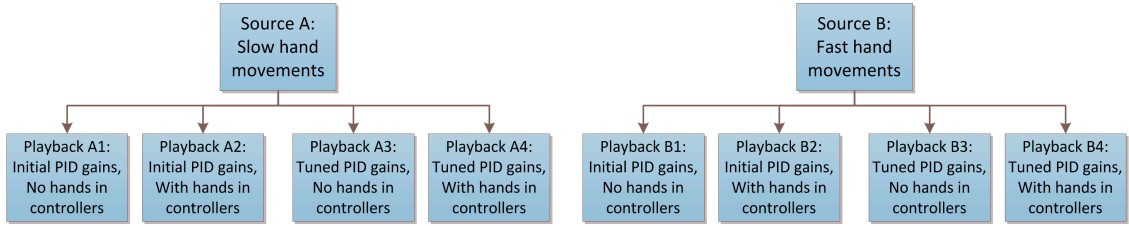

**Figure 7.** A diagram of the 10 recordings that were made to evaluate the system. A human operator moved the hand controllers for the top two source boxes, whereas the proportional–integral–derivative (PID) control system played back the corresponding source recordings for the bottom boxes.

The first recording in each group, called the "source" recording, captured the movements of the left hand controller while it was moved by a human operator. The human operator made a wide range of movements along the *x*-, *y*-, and *z*-axes of the hand controller to exercise all its joints.

The remaining 4 recordings in each group were generated by using the system to play back the data from the source recording. In other words, the left hand controller was moved automatically by the system's PID controllers according to the pre-recorded movements of the source recording. For the first 2 of the 4 recordings, the PID controllers used the original PID parameters without our tuning ($K_p$, $K_i$, and $K_d$). For the last 2 of the 4 recordings, our new PID parameters were used for the PID controllers.

For each set of PID parameters, the 2 recordings were made as follows. During the first recording, called the "no hands" recording, the system automatically moved the left hand controller while the

human operator had no interaction with the system. During the second recording, called the "with hands" recording, the human operator gently placed his hand in the left hand controller and allowed the system to guide his hand as the PID controller moved the hand controller. In other words, the user (one of the authors) was instructed to only passively hold the controllers. He tried not impart any active force to change the direction of the controller. This recording was meant to emulate a training application in which the system moves the operator's hand to assist in learning.

To help ensure a correct analysis of system accuracy, two different tracking techniques were used to measure the pose (position and orientation) of the hand controller during the recordings. The primary tracking technique used the hand controller's internal feedback system, which consists of encoders at each joint. Using a Denavit–Hartenberg model of the hand controller and standard forward kinematics equations [27], the pose of the end of the hand controller was determined at a rate of ~100 Hz.

The secondary tracking technique used AprilTags [28], a marker-based optical tracking technique. A marker was placed on the hand controller, and its pose was tracked using a calibrated camera (Figure 8). The placement of the marker and the camera were kept consistent for all recordings. As long as the hand controller was not moved too quickly or placed outside of the camera's field of view, the hand controller could be tracked. This tracking technique provided an external, independent measure of the hand controller's pose at a rate of ~30 Hz.

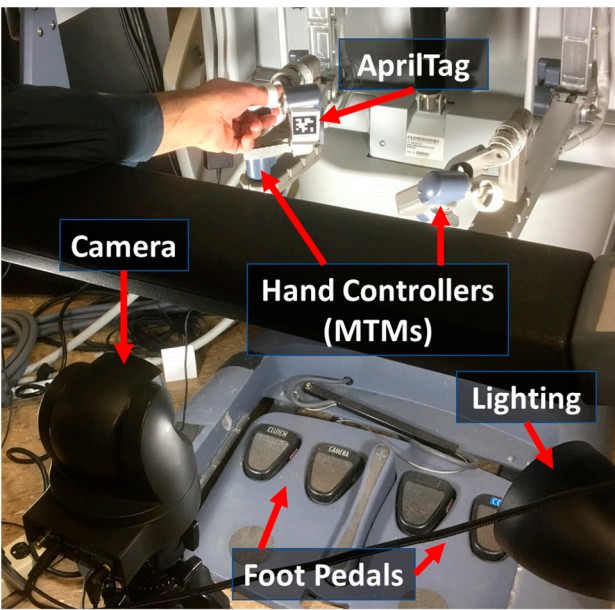

**Figure 8.** Experimental setup for evaluation of system accuracy. Camera-based tags are used on the hand controllers to capture the movement of the controllers during recording and playback sessions.

3.4.2. Processing of Test Data

The raw data collected for each of the 10 recordings consisted of the position ($x$, $y$, and $z$) and orientation of the hand controller along with a timestamp for each data point. The same type of data was available from both the internal kinematic feedback as well as the external optical tracking system, although each had a different base coordinate frame. For the analyses reported in this paper, we focused on the position data.

The recordings of the "source" movements had to occur before the system could play back the movements ("no hands" and "with hands"). Consequently, the timestamps of the recorded data from the played-back movements were shifted (delayed) relative to the recorded source data. However, our goal was to directly compare the source data with the playback data. To compensate for the differences in timestamps, we used mathematical optimization to shift the timestamps of the playback

data until the difference/error between the source and corresponding playback data was minimized. For each playback recording, a single timestamp shift was computed and applied to the entire recording, which effectively made the playback recording start at the same time as the source recording. No parts of the recording were sped up or slowed down, which would have affected the error.

The position data was processed in several ways to make it suitable for analysis. First, a median filter with a window size of 3 was applied to each axis of the data to reduce noise. Next, we used linear interpolation to resample the kinematic and optical data to a common set of time values. This was necessary due to the different sampling rates/times of the two tracking techniques. Finally, a transformation was calculated and applied to translate the optical tracking data into the same base coordinate frame as the kinematic tracking data. This is described below.

The optical tracking system returned the pose of the marker affixed to the hand controlled with respect to the camera. Both the orientation and position of the optical data had to be modified to match the base coordinate frame of the kinematic data. In terms of orientation, the optical marker was positioned on a flat surface of the hand controller in a manner that aligned with the coordinate axes of the kinematic data. Only a few simple $90°$ rotations were needed to align the orientations of the two data sets. In terms of position, the camera was located far from the origin of the base of the kinematic model. Consequently, an offset in each of the three axes ($x$, $y$, and $z$) had to be found and added to the optical position data. To find these offsets, we used a mathematical optimization routine that searched for offset values that minimized the average Euclidean distance between corresponding kinematic and optical data points. The $x$-offset, $y$-offset, and $z$-offset were uniformly applied to each optical recording, effectively changing the origin of the optical data's coordinate system without distorting the positions in a manner that would affect the error. Also note that the origin of the hand controller's coordinate system is generally arbitrary on the da Vinci because the da Vinci's clutching ability allows the origin to be freely repositioned by the user at any time.

The process of acquiring an image from the camera and locating the AprilTag also introduced a small delay relative to the tracking based on the hand controller's internal feedback. To compensate for this, we modified the mathematical optimization described above to also shift the timestamps of the optical data to minimize the distance between corresponding kinematic and optical data. For each optical recording, a single timestamp shift was computed and applied to the entire recording, which effectively made the optical recording start at the same time as the kinematic recording. No parts of the recording were sped up or slowed down, which would have affected the error.

### 3.4.3. Analysis of Test Data

After the data was processed, we analyzed and compared the various recordings and tracking techniques. In particular, we did the following:

- We computed the differences between a source recording and the recordings of the system while it played back the movements of the source recording. This included playback without a user touching the hand controllers and playback while a user gently placed his hand in the hand controller and allowed the system to guide his hand.
- We compared the performance of the playback system before and after tuning of the system's PID parameters.
- We evaluated how well the system handled slow movements as opposed to faster movements of the hand controller.
- We compared the data provided by the system's internal kinematic feedback to the data of an external optical tracking system.

Our evaluation included the direct comparison of the individual position components ($x$, $y$, and $z$) of different recordings as well as the computation of the Euclidean distance between the hand controller's position in different recordings. In some cases, we also calculated the approximate velocity

of the hand controller by dividing the change in position between two adjacent samples by the amount of time between them.

## 4. Results

In this section, we present our evaluation of the test recordings that were made using the proposed recording and playback system. We recorded data both intrinsically (from the internal joint encoder values) and using an independent external tracking system (optical tracking). In addition, we present data before and after PID optimization to show the importance of proper tuning of these parameters for the hand controller motors.

### 4.1. Overall Assessment of Playback Accuracy

The following data shows how well the proposed system was able to play back previously recorded movements of the hand controller. The data is from the system's internal kinematic feedback for the left hand controller only. The hand controller was moved slowly for this analysis, and tuned PID parameters were used for the system's motor controllers. As described previously, the data was played back under two different conditions: with the user's hands in the controllers being guided by the system, and with no hands in the controllers.

As can be seen from Figure 9, the position of the hand controllers upon playback matched the source (recorded data) quite closely. The average distance (error) between the source and the playback was less than 4 mm in both playback conditions ("with hands" and "no hands"). Most of the error occurred along the *z*-axis (this might be due to aging sensors or uncompensated gravity vector issues). The Euclidian distance between the source data and the replayed "no hands" data had a minimum of 0.49 mm, maximum of 5.62 mm, mean of 3.59 mm, and a standard deviation of 0.880771 mm. The distance between the source data and the replayed "with hands" data had a minimum of 0.46 mm, maximum of 6.37 mm, mean of 3.85 mm, and a standard deviation of 1.05 mm.

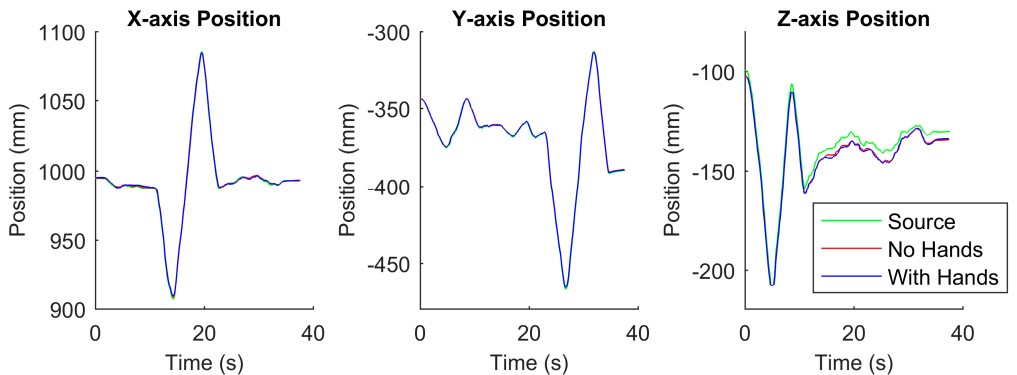

**Figure 9.** Comparison between the source data (during recording) and the replayed data (both without and with hands in the hand controllers) in terms of the *x*-, *y*-, and *z*-axis positions of the endpoint of the left hand controller. This position data was computed using joint feedback and a kinematic model of the hand controller.

### 4.2. Analysis of Error in Playback

#### 4.2.1. Results of PID tuning

To highlight the importance of properly tuning the PID controllers, we collected data on how accurately the system positioned the hand controller, with and without tuning of the PID controllers, for both slow movements and fast movements of the hand controller. Table 1 shows the initial PID values of the hand controllers (taken from [26] as described in Section 3.3.1).

Table 2 shows the resulting gains of the PID controllers after tuning. We changed the gains for four out of the seven PID controllers of each hand controller. Some of the changes were quite large,

including the gains for the outer yaw controller, shoulder pitch controller, elbow pitch controller, and wrist pitch controller. We added integral gain to the wrist pitch controller of both arms as well as to the wrist roll controller.

We note that the gains for the left elbow pitch controller are different from those for the right elbow pitch controller. These two asymmetrical PID controller values are likely due to slight differences in the mechanical parts of the two arms.

The performances of the PID controllers before and after the tuning effort are shown graphically in Figures 10 and 11. The performance was measured as the Euclidean distance between the source recording and a replayed recording ("no hands" or "with hands") for the left hand controller only. As seen from the graphs, tuning the parameters improved the accuracy of the replayed position of the hand controller. The improved controller performances are described numerically in Tables 3 and 4.

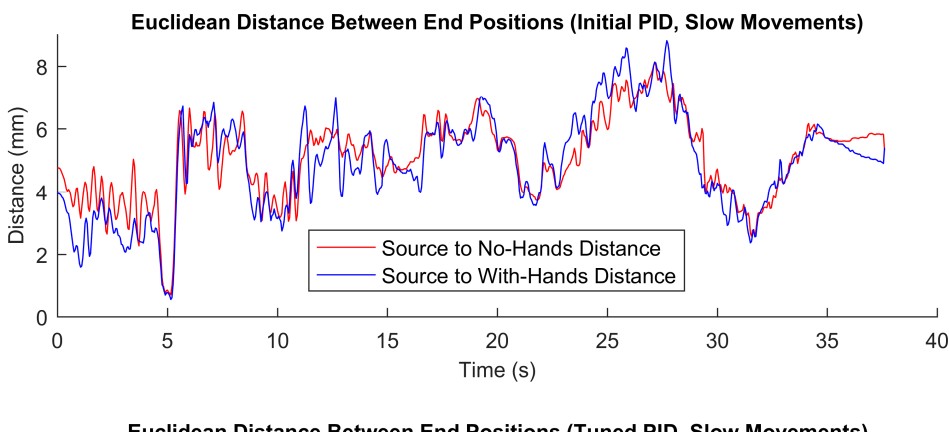

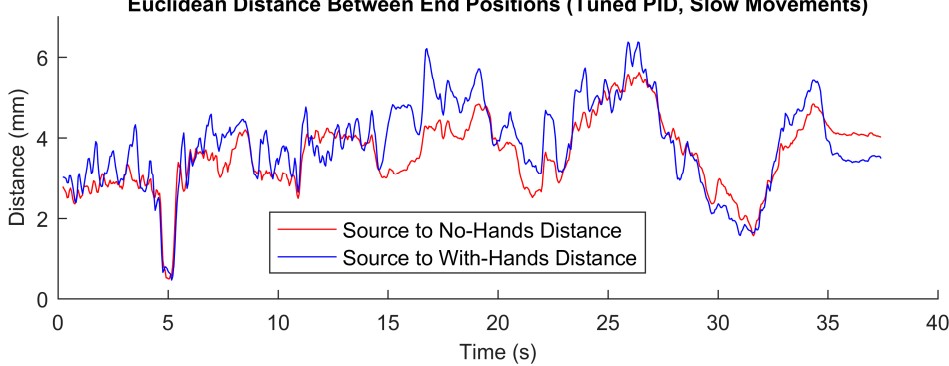

**Figure 10.** Distances between the source data (during recording) and the replayed data (both without and with hands in the hand controllers) in terms of the end position of the left hand controller during slow movements. The top graph is based on playback using the initial PID parameters (without our tuning), whereas the bottom graph is based on playback using tuned PID parameters.

**Table 3.** Playback accuracy when replaying the source recording using the PID controller for the slow movement case (recordings A1–A4 in Figure 7). The left half of table is with the initial PID parameters, and the right side is with tuned PID parameters. The top half of the table is for the no-hands condition, whereas the bottom half is with hands lightly gripping the controllers.

| | Error for Initial PID Parameters (mm) | | | | Error for Tuned PID Parameters (mm) | | | |
|---|---|---|---|---|---|---|---|---|
| | Min. | Max. | Mean | Std. Dev. | Min. | Max. | Mean | Std. Dev. |
| **Source to No Hands** | 0.69 | 8.14 | 5.07 | 1.28 | 0.49 | 5.62 | 3.59 | 0.88 |
| **Source to With Hands** | 0.54 | 8.81 | 4.93 | 1.56 | 0.46 | 6.37 | 3.85 | 1.06 |

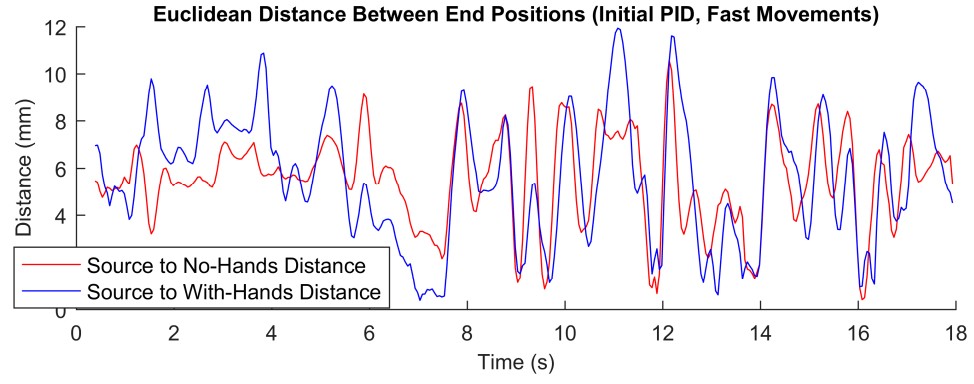

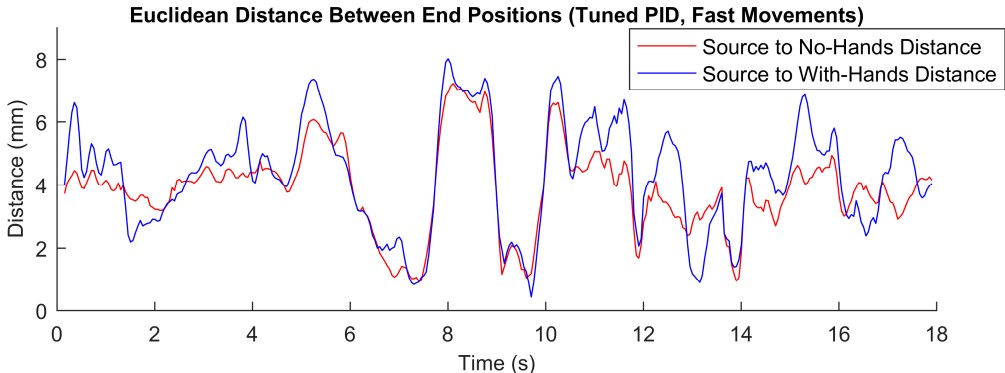

**Figure 11.** Distances between the source data (during recording) and the replayed data (both without and with hands in the hand controllers) in terms of the end position of the left hand controller during faster movements. The top graph is based on playback using the initial PID parameters (without our tuning), whereas the bottom graph is based on playback using tuned PID parameters.

**Table 4.** Playback accuracy when replaying the source recording using the PID controller for the fast movement case (recordings B1–B4 in Figure 7). The left half of table is with the initial PID parameters, and the right side is with tuned PID parameters. The top half of the table is for the no-hands condition, whereas the bottom half is with hands lightly gripping the controllers.

| | Error for Initial PID Parameters (mm) | | | | Error for Tuned PID Parameters (mm) | | | |
|---|---|---|---|---|---|---|---|---|
| | Min. | Max. | Mean | Std. Dev. | Min. | Max. | Mean | Std. Dev. |
| **Source to No Hands** | 0.41 | 10.57 | 5.51 | 1.92 | 0.93 | 7.22 | 3.87 | 1.35 |
| **Source to With Hands** | 0.38 | 11.94 | 5.60 | 2.69 | 0.43 | 8.01 | 4.29 | 1.69 |

### 4.2.2. Comparison of Different Speeds of Hand Controller Motions

The speed of the hand controller movement is a very important measure to consider. When the controller was moved very slowly, the error values (as seen from Figure 10) were generally smaller than when the controllers were moved faster (as seen from Figure 11). For fast movements (with hands in the controller), the maximum observed error was nearly 12 mm when the parameters were not tuned properly. As seen in Figure 11, with the parameters tuned properly, the error profile was much better, with the maximum error being around 8 mm. PID tuning is much more important when there are higher velocities and more dynamic behaviors involved.

The results indicate that with proper PID tuning and sufficient data recording resolution, the accuracy for the worst case (fast movements with hands in the controllers) of the playback data had an average error of 4.29 mm with a standard deviation of 1.69 mm (as seen in Table 4). The best case (for very slow movements with no hands) had an average error of 3.59 mm with a standard deviation of 0.88 mm (as shown in Table 3).

### 4.2.3. External Verification of Tracking

The primary tracking method we employed involved using recorded encoder values and a kinematic model to verify the position of end-effector. This method is inherently self-referential. In order to verify our measurements, we employed an external method of data collection to ensure that our tracking method for the hand controller is valid. We have found that an optical data collection method largely agrees with the method of using kinematic data and a robot model to derive the location of the controllers. As seen from Figure 12 and Table 5, the average error between the two methods is 4.9 mm, and the maximum error is only 16.1 mm. The maximum error represents less than 4% error when compared to the maximum movement of the hand controllers. We therefore accept the kinematic method we used to analyze our playback results.

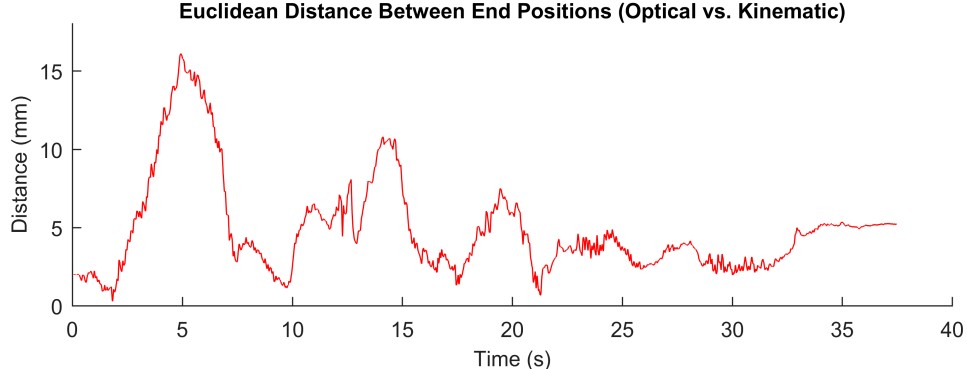

**Figure 12.** Distance between left hand controller positions measured using (1) a camera and an optically tracked marker (an AprilTag) and (2) using joint feedback and a kinematic model of the hand controller.

**Table 5.** Statistics of the distance (in mm) between left hand controller positions measured using an optical technique and using an encoder/robot-model-based technique.

| Min. | Max. | Mean | Std. Dev. |
|------|------|------|-----------|
| 0.29 | 16.11 | 4.88 | 3.11 |

## 5. Discussion

One of the central questions regarding the recording and playback system proposed here is of accuracy of playback. Although the operating surgeon can tolerate error in the robot tools, due to the visual feedback loop from his/her own movements, we think that having high accuracy for learning delicate movements may be important for some training applications of our system. We have shown that to reliably play back a recorded movement, it is very important to accurately tune the PID controllers of the system. We have also shown that this is especially important when using higher hand-controller velocities.

Our playback system could be an adjunct to traditional models of training. A key feature of our system is that major/complex operations, in their entirety, can be kinesthetically felt and watched multiple times. The system will provide numerous beneficial features. First, the playback can be slowed down at critical points to learn or evaluate the maneuvers in detail. Second, the force feedback parameters could be relaxed in such a way that it would allow the trainee to have more control over their movements (like training wheels). Third, it would be possible to add visual annotations and audio narration that would allow the trainee to be immersed in the surgery and experience knowledge-based learning along with physical movement-based learning. It is envisioned that a novice surgeon would more easily, quickly, and accurately learn the requisite kinesthetic maneuvers, proper techniques, and detailed knowledge about diverse types of surgeries from recordings of an experienced instructor. The proposed surgical data set and tools created herein could be leveraged to

potentially enhance training for surgeons. This work complements our work in task analysis based on surgical datasets [18,20]. In addition, these recordings could be ubiquitously shared beyond the local hospitals and be preserved for distribution to future generations of surgeons.

The central question related to this immersive training system is the level of efficacy that can be expected with this form of training. Can this system improve training? To answer this, a long-term study that compares different forms of training will be needed. We conjecture that to really learn complex material, the subject cannot just be a passive observer; he/she must be engaged in the activity in some way [29]. We envision a system where the user is graded during training for his/her level of engagement. For instance, as the operation is being played back, the user could be asked to manually control the gripper joints to mimic the expert movements. This way, the user must actively engage and learn to mimic and kinesthetically follow the recording during playback. A system that assesses how well the user followed the expert could be devised to gauge the level of expertise. The recorded data set, along with annotations, narrations, and software search/analysis tools could be provided to teaching and training institutions as tangible products.

In terms of training sets for deep learning methods, this data could be used in conjunction with more heuristic-based approaches to potentially learn different surgical skills such as suturing and knot tying [21]. Surgical recordings need also be decomposed to primitive motions and subtasks to be useful for learning algorithms and automation [18,20].

In addition, there are several examples of camera automation in the surgical domain [30]. Most of these systems are simplistic and only give importance to simple inputs like tool tip position or gaze point. Little to no information from actual surgical case data is used to derive behaviors. There are nuances that an expert camera operator will follow that might not be able to be captured with the simple systems that are described. The surgical recording system described here can potentially provide an extensive database from which an intelligent system can learn different behaviors.

## 6. Conclusions

In this paper, we have shown the development details of a recording and playback system for a da Vinci Standard Surgical System and research kit. The envisioned use of such a system could be for (1) training a surgeon using pre-recorded and immersive data, (2) providing data for machine learning and other applications, and (3) serving as a "black box" system (akin to airplane flight data recorders) to understand what went wrong during surgical cases. To be sure that the recorded data is of high-enough fidelity for these complex applications, we have also performed a detailed comparison of recorded vs. playback data. The system could be used by researchers to validate if augmented feedback could produce better training results or to inform the industry on how best to develop such a system.

The error between playback and recorded data was on average (across all the conditions tested) about 3–4 mm. This is reasonable when considering the kinematic chains involved and potential joint measurement errors. We recognized that certain types of very delicate surgeries may require even higher accuracy for the playback. Perhaps special hardware and high-end real-time computing techniques could reach higher levels of accuracy. A long-term study which compares the immersive training system proposed herein and traditional training methods would be needed. This future study would need to evaluate the impact on actual surgical training with many residents and over an extended period of time.

Finally, we propose that such a system could be used in a variety of application domains. Any robotic teleoperation domain (military, space, medical, etc.) used to control robots at remote locations could benefit from such data for both training and automation. In addition, areas like stroke rehabilitation where a patient needs to move his/her arm in stages could benefit from such a system (e.g., to enhance mirror therapy [31]). The developed recording and playback system is the first step towards advanced future systems that will have important benefits in surgery and many other fields.

For a short video of the recording and playback system in action, please see the following link: https://www.youtube.com/watch?v=btgeu8B_qdQ.

For access to the recording and playback software (DVRK_RecordAndPlayback) discussed in this paper, please see the following link: https://github.com/careslab/DVRK_RecordAndPlayback.

**Author Contributions:** A.P. co-directed the project, wrote the manuscript, assisted with data analysis, and helped with software design and development. S.E. assisted with software design and development, helped with tuning of parameters, and reviewed the manuscript. H.Y. assisted with the tuning of the controllers, wrote the controls aspects of the manuscript, and reviewed the manuscript. M.N. assisted with the machine learning and PID aspects of the project and reviewed the manuscript. L.A.R. co-directed the project, wrote the manuscript, assisted with data analysis, and helped with software design and development.

**Funding:** The US Department of Veterans Affairs National Center for Patient Safety provided funding under grant "NCPS Robotic Operations Task Excursion Analysis" (VA701-15-Q-O179/2VHF).

**Acknowledgments:** We wish to thank the Henry Ford Health System for donating a da Vinci Standard to the lab. We would also like to express appreciation to Shiwen Xie for his assistance with tuning of the da Vinci controllers.

**Conflicts of Interest:** The authors declare no conflicts of interest.

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
