# Peer review of "A Robotic Recording and Playback Platform for Training Surgeons and Learning Autonomous Behaviors Using the da Vinci Surgical System"

_robotics, doi:10.3390/robotics8010009_

Reviewer 1 Report

This paper presents a "Recording and Playback Platform" for the da Vinci Research Kit (DVRK). The benefits of recording data for training and machine learning are well known. The primary challenge, however, is not in the recording system, but rather in the access to realistic (e.g., clinical) data, especially given patient privacy concerns. This paper does not solve these problems; first, the DVRK is a research system and, at best, can provide data from phantom or cadaver experiments (or maybe animal surgeries in some cases). Second, the authors use an existing data collection system (ROS bags) and thus it would be difficult to claim a contribution in this area.

Instead, the paper seems to be primarily about the playback capability of the system and, mostly, about the tuning of the joint PID controllers to achieve better position tracking performance on the MTMs. It is not obvious whether this playback capability is useful for training and even less likely that it would be useful for machine learning. The paper would be strengthened by some evidence that it would be useful for at least training. I could imagine some possible benefits. For example, a trainee surgeon could learn how to better manage the limited workspace of the MTMs (e.g., when to clutch). I notice that Figure 1 indicates an "actuated clutch feedback", though the authors do not describe it further. The clutch (foot pedals) on a da Vinci are passive, so an actuated version would require some modification to the system. If the authors have already made this modification, it would be good to describe it. My guess is that this has not yet been done, since it is not shown in the YouTube video.

Second, even if playback is useful, does it need to be accurate? Note that the da Vinci is not an accurate robot, but that generally does not matter because surgeons close the loop based on visual feedback. In the Discussion, the authors state that "One of the central questions regarding the recording and playback system proposed here is of accuracy of playback". But, is this really true? It may not matter whether the position tracking error is 4 mm, 8 mm or 12 mm. To some extent, it would depend on what is being learned. The example I mentioned above (clutching to manage the workspace) likely can tolerate large position errors. Trying to learn the motions required for suturing may require better accuracy, but that is an open question. Furthermore, it would be more difficult to learn these complex motions because the MTM gripper is not actuated (i.e., you cannot playback the gripper motion). Fundamentally, one problem is that there is no stated requirement for playback accuracy, so it is not clear whether the reported accuracy improvement is necessary or sufficient.

The paper also suffers from a lack of details. For example, how was PID tuning accomplished?  Was it based on step responses, ramp responses, or some other test input?  What metrics were important -- rise time, settling time, overshoot, etc.? It is also possible that the improvement is due to using gains tuned for the specific system, rather than using the default PID gains used for all DVRKs. As the data shows, the optimal PID gain may be different even between arms on the same system. The evaluation results are not particularly significant as the data capture range is short, only 30 seconds, and the mean error for the with-hand result is within a standard deviation of the default PID gain. In any case, it would have been nice to see demonstrations that the lower error leads to better learning in trainees or machine learning algorithms (though I do not see the relevance of playback to machine learning), as that is the stated the goal of the study.

Another missing detail concerns the test motions. What are the characteristics of very-slow movements and fast. Were there stops? Direction changes? In section 3.4.1, it would help to provide a reference for typical speeds the da Vinci is operated at. Would it be possible to show a profile of how the arm was moved, alongside Figures 10 and 11, so the errors can be correlated to actions? Please also show the results for slow speeds, as Table 3 does for fast speeds.

The authors used an external optical tracker to measure the accuracy of MTM motion. This looks like a mono camera tracking an AR tag, so accuracy is probably not very good. The authors found a mean difference of about 5 mm between the optical tracker and the robot measurements, which justified their use of the robot kinematics. This is not a major result, as they are using an optical tracker with unknown (or at least unspecified) accuracy to verify that the robot is accurate to about 5 mm. How much of that 5 mm is due to the robot, and how much to the tracking system? A more interesting result would be to use an accurate optical tracking system to actually measure, with confidence, the accuracy of the robot kinematics.

In summary, this paper proposes a tool capture and playback data for robotic surgery. This is a crucial step in expanding the use of robotic surgery, both for training novices and incorporating more context-aware aid from the robot. Unfortunately, the paper does not sufficiently address the proposed problem statement to record and play back video and kinematics for training purposes. It uses a standard tool, rosbag, to capture data and shows that the PID gains can be better tuned for their system. It should discuss in more detail what the contributions of the paper are and give some indication on whether the improvements contribute towards better training.

Some minor issues:

Line 22: "Surgery is complex process" -- missing word "a"
Line 104: "In addition, There are" -- remote capitalization on "There"
Line 211: "from one or more the PID" -> "from one or more of the PID"
Line 353: text refers to "section 0"
Figure 11: caption states that the computed velocities are shown in green, but I do not see any green in the plot.
Line 398: should it be fast movement with hands? That would correspond better to Table 3.
Line 444: "A system that assess" --> "A system that assesses"
Line 456: "was on average (across all the conditions tested) was about 3-4 mm" -- redundant "was"

Reviewer 2 Report

The authors present a novel approach towards assistive surgery and surgical training, by recording and playing back real-time movements of surgeons. Although the idea of recording expert surgeon gestures is not new (see refs [1, 2, 3] below), the thought of playing it back accurately to recreate the motions on the da Vinci controllers is interesting. The article is quite well-written. I encourage the authors to improve the article to better position their contributions. My comments are below:

(1) The authors begin the article with a broad base of the possibilities with the idea of record-n-playback, including audio, video, AR, etc. This is understandable, but also builds up expectations in the readers to expect an investigation in those aspects. For instance, the authors spend a whole subsection of related work on machine learning, which has no connection with the trials that the authors present. The authors do not explore the aspects of feeding back audio or AR fixtures to the user in this paper. This part is also not evident in the video that the authors show.

The article's main contribution (the authors can correct me if I'm wrong here) is the tuning of PID parameters on the da Vinci to allow accurate playback of the motions of the controllers. The authors need to re-position the introduction and the related work sections, and introduce a 'contributions' part to tell the reader.

(2) The authors should update the related work section with refs below, and similar articles which do cover da Vinci gesture recordings.

(3) While recording the video in ROS why was the rate exactly 29.97 Hz? Was it supposed to be 30 Hz, and was lower?

(4) Could the authors explain better how the time-stamp matching was done between the 'source' data and 'playback' data? The point about mathematical optimization is mentioned in several places through the paper - it would be helpful if the authors are more specific in this. Similar is the case with transformation between the optical tracking and the kinematic tracking. How was it ensured that the shifting of time-stamp and re-sampling did not impact the error between the two methods?

(5) The authors mention that the 'with hands' data was collected with the user 'gently' placing the hand in the controllers. How was the 'gentleness' ensured? Did it impact the tracking error? Do the authors have multiple trials from which to discern the impact of 'gentleness' or lack of it?

(6) In section 3.3.2-Tuning of PID, on Line 215, the authors mention 'Table 1' for the final tuning gains. I think they mean Table 2.

(7) Does equation (1) for the PID control come from da Vinci or have the authors introduced the 'K_n" factor there? If the authors have introduced it, then Table 1 should not have "K_n" value in it.

(8) Figures 10 and 11 do not show the 'computed velocity' in green that the authors mention in the captions.

(9) The authors should present the results tables clearly for the conditions that they mention, i.e., A1 through B4. If I have understood correctly, section 4.1 talks about conditions A3 and A4 - slow motion, tuned PID, no hands and with hands. Then it is not clear what data Table 3 presents, because it does not match with section 4.1. Likewise the tables for B1-to-B4 are not presented, but only graphically shown with Figure 11.

References:

[1] Sridhar, A. N., Briggs, T. P., Kelly, J. D., & Nathan, S. (2017). Training in Robotic Surgery-an Overview. Current urology reports, 18(8), 58.

[2] Kumar, R., Jog, A., Vagvolgyi, B., Nguyen, H., Hager, G., Chen, C. C., & Yuh, D. (2011). Objective measures for longitudinal assessment of robotic surgery training. The Journal of thoracic and cardiovascular surgery, 143(3), 528-34.

[3] Sriram Garudeswaran, Sohyung Cho, Ikechukwu Ohu, and Ali K. Panahi, “Teach and Playback Training Device for Minimally Invasive Surgery,” Minimally Invasive Surgery, vol. 2018, Article ID 4815761, 8 pages, 2018.

Author Response

Round  2

Reviewer 2 Report

The authors have responded positively to the suggestions given in my earlier review. As such, I am satisfied with the improvements made to the manuscript. But the formatting of the submitted revision is not good, especially in results section with incorrect table layout. It seems like the revision for submitted in haste without completely checking it. Comments below:

(1) Line #291, is MTM an acronym? The authors should mention the expanded form before using it.

(2) Line @292, authors should check grammar of "intended to correlating to".

(3) Line #349, for "compression or expansion of time", do the authors mean "compression or expansion of recording"? Same for line #376.

(4) There are two sections 4.1, line #402 and #423. The section numbering and heading should be checked completely.

(5) I am not sure why the authors have submitted the "tracking changes on" version for this revised version. The formatting is completely off in the results section. Line #442 and on is the caption for Table 3, is that correct? Line #448 says Table 4 and Table 4? Table 3 appears after line #446 and then again after line #464. Tables 1 and 2 appear after Table 3.

(6) In section 4.1.2, the authors mention Figure 12 for error profile of tuned PID gains - I think they mean Figure 11 itself.

(7) The discussion section is quite long and can be reduced.
